# Sedative Properties of Dexmedetomidine Are Mediated Independently from Native Thalamic Hyperpolarization-Activated Cyclic Nucleotide-Gated Channel Function at Clinically Relevant Concentrations

**DOI:** 10.3390/ijms24010519

**Published:** 2022-12-28

**Authors:** Stefan Schwerin, Catharina Westphal, Claudia Klug, Gerhard Schneider, Matthias Kreuzer, Rainer Haseneder, Stephan Kratzer

**Affiliations:** Department of Anesthesiology and Intensive Care Medicine, Klinikum Rechts der Isar, Technical University of Munich School of Medicine, 81675 Munich, Germany

**Keywords:** dexmedetomidine, hcn channel, thalamus, thalamocortical relay neuron, patch-clamp, anesthesia

## Abstract

Dexmedetomidine is a selective α_2_-adrenoceptor agonist and appears to disinhibit endogenous sleep-promoting pathways, as well as to attenuate noradrenergic excitation. Recent evidence suggests that dexmedetomidine might also directly inhibit hyperpolarization-activated cyclic-nucleotide gated (HCN) channels. We analyzed the effects of dexmedetomidine on native HCN channel function in thalamocortical relay neurons of the ventrobasal complex of the thalamus from mice, performing whole-cell patch-clamp recordings. Over a clinically relevant range of concentrations (1–10 µM), the effects of dexmedetomidine were modest. At a concentration of 10 µM, dexmedetomidine significantly reduced maximal I_h_ amplitude (relative reduction: 0.86 [0.78–0.91], n = 10, and *p* = 0.021), yet changes to the half-maximal activation potential V_1/2_ occurred exclusively in the presence of the very high concentration of 100 µM (−4,7 [−7.5–−4.0] mV, n = 10, and *p* = 0.009). Coincidentally, only the very high concentration of 100 µM induced a significant deceleration of the fast component of the HCN activation time course (τ_fast_: +135.1 [+64.7–+151.3] ms, n = 10, and *p* = 0.002). With the exception of significantly increasing the membrane input resistance (starting at 10 µM), dexmedetomidine did not affect biophysical membrane properties and HCN channel-mediated parameters of neuronal excitability. Hence, the sedative qualities of dexmedetomidine and its effect on the thalamocortical network are not decisively shaped by direct inhibition of HCN channel function.

## 1. Introduction

Dexmedetomidine (DEX) is commonly used for short- and longer-term sedation of adult patients receiving treatment in intensive care units and as an anesthetic adjunct due to its generally favorable pharmacological profile, including anxiolytic, analgesic, and opioid-saving properties [1,2]. When examining the effects of DEX on neural network activity, it appears as though it induces a rather unique brain state. Electroencephalographic (EEG) recordings of patients being administered DEX are characterized by slow-wave (SW, 0.1–1 Hz) and delta (δ, 1–4 Hz) oscillations, as well as spindle activity (12–16 Hz) [3]. Despite perfunctory similarities with EEG features observable during general anesthesia with propofol, which induces profound unconsciousness, DEX shifts brain activity in a rather different, sleep-like state [4]. Patients who receive DEX for sedation, can be aroused readily, maintain the ability to communicate and cooperate properly, and further do not display some of the adverse effects generally accompanied by other hypnotic agents, such as respiratory depression [2,5].Among a long list of potential suspects, a relatively small number of key molecular targets of general anesthetics have been identified [6]. Deviating from this canonical compilation, DEX predominantly acts as a selective agonist of presynaptic alpha-2 (α_2_)-adrenergic receptors, which are located on axonal projections of glutamatergic neurons originating in the locus coeruleus (LC). By decreasing noradrenaline release in a dose-dependent manner [7], DEX results in hyperpolarization and reduced excitability of LC neurons [8]. Inhibition of LC neurons is thought to partly entail the promotion of endogenous non-rapid eye movement sleep (NREM)-promoting pathways [9] with manifold consequences on a network level [5], including the sleep-promoting disinhibition of the hypothalamic ventrolateral preoptic nucleus (VLPO), which in turn inhibits the tuberomammillary nucleus as part of the ascending monoaminergic arousal center via γ-aminobutyric acid (GABA)-ergic and galanin-ergic VLPO projections [9,10,11,12]. Of the three distinct subtypes of α_2_-adrenergic receptors identified (α_2A_, α_2B_, and α_2C_), α_2A_-receptor subtypes appear to primarily mediate sedative, hypnotic, as well as anesthesia-sparing effects in mice [13,14,15].

Further effects of DEX include attenuated excitation of arousal-related basal forebrain regions, the thalamus and cortex [16], as well as antinociception both at the spinal cord level as well as at supraspinal sites [17]. The application of sedative concentrations of DEX results in reduced thalamocortical (TC) functional connectivity, a diminished cerebral metabolic rate of glucose, as well as an attenuated cerebral blood flow, preferentially in the thalamus [18].

As the LC heavily projects both onto the thalamus and the cortex [19], the observed changes are likely due to a DEX-mediated dampening of ascending arousal pathways, resulting in hyperpolarization of TC relay neurons and thereby interrupting TC communication [20]. Hyperpolarization of TC relay neurons is an important requisite for generating and maintaining synchronized δ-oscillations within the TC network, as these neurons transition from a tonic, single-spike activity to a rhythmic, burst-firing mode [6,21]. In addition to hyperpolarization as a result of dampened ascending input, DEX might also directly modulate the intrinsic excitability of TC relay neurons. It is established that properties of thalamic oscillations and the pacemaker function of TC relay neurons are decisively shaped by the inwardly-directed, mixed cation current I_h_ [22,23], which is mediated by hyperpolarization-activated cyclic nucleotide-gated channels (HCN).

How HCN channels mediated excitability of TC relay neurons is affected by anesthetic agents has been extensively investigated over the past decades [6,24] and a number of in vitro studies demonstrate inhibition of native HCN channel function in the cases of propofol [25], pentobarbital [26], xenon [27], halothane [28,29], and sevoflurane [30]. Interestingly, recent evidence suggests that DEX might also act on heterologously expressed HCN channels and attenuate HCN channel function in neurons from dorsal root ganglia [31,32]. This is further supported by the notion that other selective α_2_-adrenoceptor agonists, including clonidine, have been shown to inhibit HCN channel function [33,34,35,36]. DEX-mediated effects on the neurophysiology of the brain might, therefore, arise from direct action on VB neurons, in addition to the effects that are elicited by the modulation of neurons originating for the LC, primarily via α_2A_-receptor subtypes [14].

Hence, we investigated whether DEX, at clinically relevant concentrations, affects native HCN channel function and HCN channel-mediated excitability of TC relay neurons located in the primarily first-order ventrobasal (VB) complex of the thalamus [37], using the patch-clamp technique in acutely prepared brain slices from mice.

## 2. Results

### 2.1. Effects of DEX on Membrane Properties of TC Neurons

Passive and active membrane biophysics were derived from current–voltage relationship recordings. Under baseline recordings, TC neurons displayed a median resting membrane potential of −58.2 (IQR: −59.7–−56.9) mV, n = 31. For all three DEX concentration groups, we did not detect a significant change in the resting membrane potential after 30 min and 45 min of DEX application, see Appendix A. The effects of DEX are presented as an absolute difference from the baseline recordings (**Δ 1 µM DEX:** −0.31 [−1.5–+1.2] mV, n = 11, *p* > 0.999; **Δ 10 µM DEX:** +0.8 [−1.9–+1.5] mV, n = 10, *p* > 0.999; and **Δ 100 µM DEX:** −0.8 [−2.6–+0.2] mV, n = 10, *p* > 0.342). In between concentration groups, there were no significant differences with regard to baseline conditions (*p* = 0.671, Kruskal–Wallis test with a posthoc Dunn’s test). A total of 40 µM of HCN channel blocker ZD7288, which has a reported half maximal inhibitory concentration (IC_50_) of 15 µM [38], induced a strong hyperpolarization of the resting membrane potential in all cells measured, which persisted after the subsequent application of 10 µM DEX (**control:** −60.9 [−63.2–−58.5] mV]; **+40 µM ZD7288:** −65.5 [−74.0–−61.0] mV; and **+10 µm DEX:** −63.5 [−72.1–−61.0] mV, n = 5). See Figure 1A.

Beginning at a concentration of 10 µM DEX, we observed a significant increase in the membrane input resistance, indicative of decreased neuronal leakiness. Under control conditions, the median input resistance was at 343.4 (252.3–365.6) MΩ, n = 31. The effects of DEX are presented as an absolute difference from the baseline recordings (**Δ 1 µM DEX:** +24.0 [−10.0–+37.2] MΩ, n = 11, *p* = 0.602; **Δ 10 µM DEX:** +27.9 [+9.6–+89.0] MΩ, n = 10, *p* = 0.011; and **Δ 100 µM DEX:** +95.3 [+73.2–+120.5] MΩ, n = 10, *p* < 0.001). A total of 40 µM ZD7288 also resulted in an increase in the input resistance in all cells recorded, which again persisted after applying 10 µM DEX (an absolute difference from the baseline **Δ +40 µM ZD7288:** +140.3 [+63.9–+340.5] MΩ and +**10 µm DEX:** +159.1 [+34.0–+378.3] MΩ, n = 5) Figure 1B. However, when analyzing changes over the whole current–voltage relationship, the increase of the input resistance in presence of 10 µM DEX was only statistically significant during hyperpolarizing current injections, in the range from −90 pA to −70 pA (an absolute change of the input resistance after 10 µM DEX at −70 pA current injection compared to the baseline: +38.2 [+19.3–+53.4] MΩ, n = 10, *p* = 0.037 and at −60 pA: +40 [+3.2–+64.9] MΩ, n = 10, *p* = 0.084).

When analyzing the effect of DEX on current–voltage relationships of recorded cells, which were determined by injecting successively depolarizing currents (10 pA increments), only the high concentration of 100 µM DEX persistently induced a significant hyperpolarization in the whole range between −90 and +350 pA, whereas 10 µM DEX only induced significant changes during strongly hyperpolarizing currents, see Figure 1C. Current–voltage relationships were not recorded in the presence of agents, which selectively prevent the occurrence of action potentials. This might be reflected by broader interquartile ranges of voltage responses obtained by injection of rather high current pulses.

Similarly, intrinsic firing properties of TC neurons were only significantly affected by a very high concentration of 100 µM DEX. Under the baseline conditions, the threshold potential for action potential firing was determined at −39.2 (−40.5–−37.6) mV. 100 µM DEX shifted this threshold toward depolarization, with no significant effects occurring at lower concentrations (**Δ 1 µM DEX:** −0.6 [−2.6–−0.2] mV, n = 11, *p* = 0.583; **Δ 10 µM DEX:** −0.4 [−2.4–+0.9] mV, n = 10, *p* = 0.850; and **Δ 100 µM DEX:** +1.3 [+1.1–+5.4] mV, n = 10, *p* = 0.013). In contrast, HCN channel blocker ZD7288 shifted the threshold potential toward hyperpolarization. The co-application of ZD7288 and 10 µM DEX did not reverse the observed hyperpolarization (control: −38.1 [−39.4–−36.7] mV; **Δ 40 µM ZD7288:** −4.8 [−7.6–−2.4] mV; and **Δ 10 µM DEX + ZD7288:** −3.7 [−7.3–−2.4] mV, n = 5), see Figure 1D. 

We further analyzed the effect of DEX on the maximum frequency of tonic action potential firing. The frequency was derived from the current injection of +190 pA, as we did not observe a further increase in maximum frequency beyond this value. Median tonic action potential firing at the baseline conditions occurred at 40.2 (36.3–44.9) Hz, n = 31. Again, significant changes were exclusively observable in the presence of the high DEX concentration, with a decrease in tonic action potential frequency (**Δ 1 µM DEX:** +2.3 [−1.8–°+5.6] Hz n = 11, *p* = 0.953; **Δ 10 µM DEX:** −0.7 [−2.2–+3.1] Hz, n = 10, *p* > 0.999; and **Δ 100 µM DEX:** −12.8 [−18.7–−9.7] Hz, n = 10, *p* < 0.001), see Figure 1E.

All experiments were conducted in the presence of 150 µM Ba^2+^ to preclude interference with K_ir_ and K_2P_ channels. On native recordings, the addition of Ba^2+^ resulted in a small depolarization of the resting membrane potential (**control:** −57.6 [−59.5–−57.4] mV vs. **Ba^2+^**: −56.2 [−58.7–−55.8] mV, n = 7, *p* = 0.016) and an increase in the input resistance (**control:** 180.6 [133.7–197.6] MΩ vs. **Ba^2+^**: 229.9 [212.5–263.2] MΩ, n = 7, *p* = 0.016), whereas there were no significant changes regarding the action potential threshold (*p* = 0.578) as well as tonic action potential firing (*p* = 0.094).

### 2.2. Influence of DEX on HCN Channel Function and I_h_ Currents

We analyzed the effects of DEX on the HCN channel-mediated, mixed cation current I_h_, as well as modalities of HCN channel function. Since the extent of HCN channel activation correlates with increasing hyperpolarization [39], the I_h_ current amplitude derived at −133 mV was considered the maximum (I_h max_) current amplitude, and I_h_ current amplitudes evoked at more depolarized values were set in relation accordingly, see Figure 2A. Both 10 µM and 100 µM DEX applications resulted in a significant reduction in I_h max_ (values are relative to the normalized baseline recordings; **1 µM DEX:** 0.86 [0.79–0.98], n = 11, *p* = 0.068; **10 µM DEX:** 0.86 [0.78–0.91], n = 10, *p* = 0.021; and **100 µM DEX:** 0.78 [0.50–0.84], n = 10, *p* < 0.001). By contrast, HCN channel blocker ZD7288 (40 µM) led to an almost complete reduction in I_h max_ in all cells recorded (values relative to the normalized baseline recordings; **40 µM ZD7288:** 0.05 [0.02–0.06], n = 5), see Figure 2B. At a membrane potential of −73 mV, reflecting more physiological conditions with only partial HCN channel activation, only 100 µM DEX induced a significant decrease (**1 µM DEX:** 0.92 [0.86–1.05], n = 11, *p* > 0.999; **10 µM DEX:** 0.90 [0.80–1.00], n = 10, *p* = 0.293; **100 µM DEX:** 0.57 [0.42–0.58], n = 10, *p* < 0.001; and **40 µM ZD7288:** 0.05 [0.03–0.14], n = 5), see Figure 2C. However, in the case of 10 µM DEX, the reduction in I_h_ current amplitude became significant beyond membrane potentials of −83 mV.

Furthermore, we evaluated whether DEX affects the voltage-dependent activation of HCN channels and open channel probability via a tail current analysis. The respective I_tail_ activation curves fitted with a Boltzmann function are shown in Figure 3A. We observed a median half-maximal activation potential (V_1/2_) of −86.6 (−85.2–−88.0) mV for TC relay neurons under baseline conditions. Only in the presence of the high concentration of 100 µM DEX was the V_1/2_ shifted toward hyperpolarization in accordance with a reduced open-channel probability (**Δ 1 µM DEX:** +0.3 [−0.9–+1.1] mV, n = 11, *p* > 0.999; **Δ 10 µM DEX:** +0.4 [−1.5–+1.7] mV, n = 10, *p* > 0.999; and **Δ 100 µM DEX:** -4.7 [−7.5–−4.0] mV, n = 10, *p* = 0.009), see Figure 3B.

In comparison to native recordings, the addition of 150 µM Ba^2+^ did not significantly change I_h max_ (**control:** 976.7 [552.6–1291.4] pA vs. **Ba^2+^**: 916.0 [516.8–1128.4] pA, n = 6, *p* = 0.844) nor V_1/2_ (**control:** −87.1 [−90.0–−83.4] mV vs. **Ba^2+^**: −86.0 [−88.6–−84.9] pA, n = 6, *p* = 0.844).

A similar pattern of only modest effects at high DEX concentrations was seen with regard to the time course of I_h_ activation, which was evaluated by fitting the I_h_ current recorded at −133 mV to a biexponential function, see Figure 3C. Absolute values of the corresponding activation time constants τ_fast_ and τ_slow_ change considerably, depending on the voltage step analyzed, as well as the duration of the current recording. Due to the limited effects observed with regard to I_h_ amplitude, we chose to apply the biexponential fitting to the maximum voltage step at −133 mV. The median value of τ_fast_ at −133 mV was 231 ms (214–273 ms; n = 31), while τ_slow_ was 1143 ms (986–1405 ms; n = 31) under control conditions. Only the fast component of HCN activation kinetics was subjected to a significant deceleration, while the slow component remained, overall, unaffected. Likewise, the observed deceleration was only significant at a high concentration of 100 µM DEX (**τ_fast_: Δ 1 µM DEX:** −4.1 [−20.3–+4.9] ms, n = 11, *p* > 0.999; **Δ 10 µM DEX:** +25.5 [+4.1–+39.6] ms, n = 10, *p* = 0.232; **Δ 100 µM DEX:** +135.1 [+64.7–+151.3], n = 10, *p* = 0.002; **τ_slow_: Δ 1 µM DEX:** +93.0 [−205.9–−198.4] ms, n = 11, *p* > 0.999; **Δ 10 µM DEX:** +163.8 [−150.4–+574.2] ms, n = 10, *p* > 0.999; and **Δ 100 µM DEX:** −208.2 [−697.5–+155.0], n = 10, *p* = 0.382), see Figure 3C.

### 2.3. Effects of DEX on TC Neuron Firing Properties

Lastly, we evaluated whether DEX affects modalities of neuronal excitability in TC relay neurons generally associated with HCN channel function. Therefore, we performed additional whole-cell current-clamp recordings. Upon injection of a hyperpolarizing current (−350 pA, 500 ms), TC neurons display an anomalous rectification of the membrane potential, termed “voltage sag”, followed by a low-threshold, calcium burst of action potential firing. Upon burst cessation, the neurons show an afterdepolarization. Both afterdepolarization and voltage sag are largely mediated by HCN channels [40], see Figure 4A. All current-clamp recordings were recorded in the presence of 150 µM Ba^2+^. In a subset of experiments, we evaluated the impact of Ba^2+^ on native current-clamp recordings. As shown in Figure 4A, Ba^2+^ unmasks an otherwise discreet voltage sag (**control:** 14.8 [5.7–18.7] mV vs. **Ba^2+^**: 58.9 [53.2–63.9], n = 7, *p* = 0.016) and significantly accelerates the median rebound burst delay (**control:** 62.5 [49.0–86.0] ms vs. **Ba^2+^**: 33.5 [29.5–44.0] ms, n = 7, *p* = 0.016). In comparison, HCN channel blocker ZD7288 completely attenuates the inwardly directed voltage rectification. Further, in two out of five cells, we were not able to induce bursting activity in the presence of ZD7288, even when increasing the current injection to −500 pA or compensating for the ZD7288-induced membrane hyperpolarization. In comparison, the effects of DEX were negligible, see Figure 4B.

The median voltage sag amplitude displayed by TC neurons under baseline conditions was 86.4 (70.7–92.9) mV (n = 31). Interestingly, even the high concentration of 100 µM DEX, which was sufficient to elicit a significant change in I_h_ amplitude, half-maximal activation potential, as well as a deceleration of the fast activation kinetics time constant, did not affect the voltage sag (**Δ 1 µM DEX:** −0.4 [−2.9–+1.3] mV, n = 11, *p* > 0.999; **Δ 10 µM DEX:** −1.0 [−4.8–+4.4] mV, n = 10, *p* = 0.753; and **Δ 100 µM DEX:** +5.1 [+0.4–+12.4], n = 10, *p* = 0.083), see Figure 4C. 

Significant changes with regard to rebound delay, the number of action potentials during rebound spike activation, as well as median burst duration exclusively occurred at the high concentration of 100 µM DEX. During control conditions, the median delay of rebound burst-spiking upon hyperpolarization was 29.0 ms (27.0–30.8 ms; n = 31), with a significant prolongation of 6.5 ms (5.1–8.9) ms (**Δ 100 µM DEX:** n = 10, *p* = 0.003; lower concentrations not shown), see Figure 4D. The median number of action potentials observed during rebound burst-firing was 9.5 (7.3–12.5) under control conditions, with a significant reduction of −6.8 (−8.8–−4.7) action potentials (**Δ 100 µM DEX:** n = 10, *p* = 0.002, see Figure 4E). Corresponding with the lower number of action potentials, the median duration of rebound bursts (under control conditions of 225.5 [126.5–399.0] ms) was only significantly reduced at the high concentration (**Δ 100 µM DEX:** −140.3 [−247.1–−72.8] ms, n = 10, *p* = 0.016), see Figure 4F.

In conclusion, our data demonstrate that DEX does not significantly decrease voltage sag amplitudes or attenuate rebound burst-firing in TC relay neurons, which are thought to be governed by an HCN conductance, at least not at clinically relevant concentrations.

Absolute values of all parameters analyzed in this study before and after DEX application are presented in Appendix A for each concentration group separately. The supplementary tables further include corresponding paired statistical analysis.

## 3. Discussion

We used patch-clamp recordings of TC relay neurons from the VB thalamic complex to investigate whether DEX can directly inhibit I_h_ or HCN channel-mediated parameters of neuronal excitability ex vivo. Our principal finding is that the effects of DEX in clinically relevant concentrations on native HCN channel function are modest.

A total of 10 µM DEX significantly reduced I_h max_ and increased the neuronal input resistance, albeit only in the range of hyperpolarizing current injections. HCN channels decisively regulate the resting membrane potential of VB TC relay neurons [41] and, therefore, inhibitory effects are to be expected primarily in the hyperpolarized membrane range. However, we did not observe hyperpolarization of the resting membrane potential and there were no significant changes in the half-maximal activation voltage within the clinically relevant concentration range. The relationship between pharmacological HCN channel inhibition, neuronal input resistance, and resting membrane potential is not always straightforward. Xenon inhibited HCN channels and increased neuronal input resistance, but did not hyperpolarize the resting membrane potential of TC relay neurons [27]. Propofol and sevoflurane both inhibit HCN channels and hyperpolarize the resting membrane potential, but concomitantly decrease neuronal input resistance, likely via GABA_A_-receptor activation [25,30,42]. With the inclusion of barium, we minimized the influence of K_ir_ and subfamilies of K_2P_ channels [40], but we cannot positively preclude the activation of further conductances. Furthermore, the reduction in I_h_ observed at −133 mV at 10 µM DEX might not be physiologically relevant enough to translate into changes in the resting membrane potential, especially as changes of I_h_ at −73 mV were not statistically significant. This is further supported by the observation that DEX did not attenuate the inwardly rectifying voltage sag, in contrast to other anesthetics [25,27,30]. As the resting membrane potential remained unaffected, all current-clamp recordings were performed at the resting membrane potential rather than different fixed voltages. In comparison, HCN channel blocker ZD7288 hyperpolarized the resting membrane potential and increased the neuronal input resistance of all cells recorded, while simultaneously diminishing I_h_ over the whole voltage range. Furthermore, our findings with regard to the very high concentration of 100 µM DEX should be interpreted with caution, as we cannot positively exclude unspecific receptor interactions or the occurrence of other conductances.

An obvious concern with respect to in vitro studies relates to the question of whether the applied concentrations of a pharmacological agent are indeed clinically relevant. A recent study [43] states that the clinically relevant DEX concentration range lies from 1 to 2.5 µM in the case of rats. A total of 5 µM was already regarded as the upper end of the scale. Incidentally noted, this study also demonstrated that DEX appears to be non-cytotoxic and did not induce cell death in cultured neonatal rat neurons. Consistent herewith, the half-maximal effective concentration (EC_50_) of DEX required for sedation and analgesia in rats was determined to be at 1 µM [44]. Some sources assume considerably lower dosages (0.001–1 µM DEX) to be relevant [45]. However, mice appear to generally require a higher minimum DEX dose to induce sedation [9]. Therefore, in vitro studies investigating the effect of DEX in mice neurons tend to apply slightly higher concentrations. In mice dorsal root ganglion neurons, DEX reduced capsaicin-induced intracellular calcium concentrations in a dose-dependent manner between 1 and 50 µM DEX [46]. In vitro studies explicitly investigating the effect of DEX on HCN channel function have also applied mostly similar concentrations (0.1–10 µM DEX [31,32], 0.1–1 µM DEX [47], and 10 nM–6 µM DEX [48]). We, therefore, consider the applied concentrations of 1 and 10 µM DEX in this study to be clinically relevant with regard to a hypothesized effect on HCN channels, while 100 µM certainly represents a supratherapeutic concentration. It is also necessary to ensure that DEX can sufficiently penetrate the ex vivo brain slice preparation, especially when comparing our results to in vitro studies. Two studies using brain slice preparations mention that DEX, as well as the pharmacologically similar drug clonidine, reduced I_h_ amplitude already within 5 min after application [35,48]. Our experiments were conducted over the comparatively long timespan of 45 min (the results presented here reflect the comparison between baseline measurements and DEX application after 45 min) and any DEX-induced changes should have occurred within this interval. We further conducted recordings after 30 min, but there were no significant differences when compared to 45 min. Additionally, the previously reported DEX-induced inhibition of I_h_ and HCN channel function was characterized as non-recoverable even after a prolonged washout [48].

The goal of this study was to investigate whether DEX can directly inhibit HCN channel function. One indication of this assumption was based on the observation that clonidine, another α_2_-adrenoceptor agonist, has been shown to directly inhibit HCN2 and HCN4 (as well as HCN1, albeit with less sensitivity) channels in transfected HEK293 cells and block I_h_ in isolated sinoatrial node pacemaker cells. Comparable to the HCN blocking agent ZD7288, clonidine also shifted V_1/2_ toward more hyperpolarized potentials [34]. In dopaminergic neurons of the ventral tegmental area, clonidine decreased I_h_ amplitude and slowed its rate of activation independently from cAMP via the activation of protein kinase C [35]. Furthermore, clonidine has hitherto been shown to inhibit HCN-mediated I_h_ in further neuronal cell types, including rat trigeminal root ganglion neurons [49], rat hypoglossal motoneurons [33], and rat pyramidal neurons in the prefrontal cortex [50].

So far, our knowledge about the effects of DEX on HCN channels is still limited. There is indirect evidence from a study using an in vivo model of peripheral nerve blockade in rats [51]. Here, adding DEX to the local anesthetic ropivacaine prolonged analgesia compared to analgesia with ropivacaine alone. Prima facie, our findings are contrasted by the results of several electrophysiological in vitro studies. One group investigated the effects of DEX on HCN channel function in view of neuropathic pain and analgesia, complementary to behavioral tests [31,32]. The authors report that DEX, in concentrations ranging from 0.1 to 10 µM, inhibits both mHCN1 and mHCN2 channels expressed in HEK293 cells. Significant differences between HCN1 and HCN2 in relative DEX inhibition were not found [31]. In a follow-up study, the same authors report patch-clamp recordings from HEK293 cells expressing HCN1 and HCN2 [32]. Another study [47] analyzed the effect of DEX on cultured rat pituitary tumor cells (GH_3_). DEX induced a concentration-dependent reduction in the amplitude of I_h_ (IC_50_ of 1.21 µM) and a substantial hyperpolarizing shift in the voltage-dependent activation course of I_h_ (−10 mV). Further, DEX attenuated I_h_ in pheochromocytoma PC12 cells and slowed its activation time course. Endocrine and neuroendocrine cells, including GH_3_ cells, intrinsically express isoforms HCN2 and HCN3 [52]. Lastly, a study investigated the effect of DEX on hypothalamic paraventricular nucleus (PVN) neurons from rat brain slice preparations [48]. DEX reduced the I_h_ amplitude irreversibly and the V_1/2_ was shifted toward hyperpolarization by −8 mV This effect was only marginally dependent on intracellular cAMP. Rat PVN neurons have been shown to express mRNA of HCN1, HCN2, and HCN3, but not HCN4 [53]. 

Furthermore, DEX dose-dependently hyperpolarized PVN neurons and decreased input resistance, but this hyperpolarization was blocked by pretreatment with 0.3 mM Ba^2+^ [48]. Therefore, the hyperpolarizing effect of DEX is likely facilitated by K_ir_ and certain potassium K_2P_ channels [54]. This could explain the absence of hyperpolarization in our recordings, as all our experiments were conducted in presence of Ba^2+^. In contrast with the reported decrease in input resistance in the study by Shirasaka et al. [48], we observed a significant increase in membrane input resistance beginning at 10 µM. This increase in input resistance would be, in principle, consistent with HCN channel inhibition [39]. However, the observed increase in the neuronal input resistance at 10 µM DEX was only significant during strongly hyperpolarizing current injections. We previously reported a similar effect in the case of xenon [27]. Many anesthetic agents, such as propofol and sevoflurane, decrease neuronal input resistance, probably via concomitant allosteric activation of GABA_A_-receptors [25,30,55]. This assumption is further supported by the observation that xenon both inhibits I_h_ and increases input resistance [27] while eliciting only minimal GABAergic responses [56]. However, propofol, sevoflurane, and xenon clearly attenuated native HCN channel function in native TC relay neurons.

As a matter of principle, recordings of HCN-mediated I_h_ are known to be strongly sensitive to changes in experimental conditions, including the ionic composition of extra- and intra-cellular solutions, temperature, pH, and patch-clamp configuration [39]. In contrast to heterologously expressed HCN channel isoforms or cultured cells, cells from ex vivo brain slice preparations are further integrated into complex modulatory networks. Accordingly, ex vivo preparations have been an integral and well-appreciated component of anesthesia research over the last decades [57]. However, we previously demonstrated pharmacological inhibition of HCN channel function in TC neurons, both in the cases of xenon and sevoflurane, using the same method and brain slice preparation [27,30]. Instead, we would like to argue that an important aspect of the question, of why our findings diverge from aforementioned studies [31,32,47,48], might relate to HCN channel subunit composition. The HCN family encompasses four different homologous subunits (HCN1–HCN4). These subunits assemble as homo- and hetero-tetramers [58] and differ with respect to their pharmacological profile and biophysical properties [58,59,60]. These quantitative differences include the degree of cAMP regulation, voltage-dependence of activation, and steady-state properties, as well as activation time constants [39]. HCN channels and corresponding I_h_ currents are present throughout the mammalian nervous system, but the distribution of specific HCN isoforms and their combinations display a more heterogeneous pattern [39]. TC relay neurons of the VB complex predominantly express HCN2 and HCN4 isoforms [61,62] and, correspondingly, studies demonstrated that a proper I_h_ conductance in TC relay neurons is largely dependent on HCN2 [63] and also HCN4 [64]. Importantly, subunit selective modulation of HCN channels by anesthetic and other pharmacological agents has been demonstrated before [28,65,66]. With regard to their activation kinetics, HCN subunits act quite differently [39]. As the opening kinetics accelerate with increasing hyperpolarization [59], specific τ values depend on the hyperpolarization step they were derived from. HCN2 displays intermediate opening kinetics compared to other isoforms and its τ-value ranges from 150 ms to 1 s (−140 mV to −70 mV), whereas the pronouncedly slower HCN4 isoform ranges from several hundred milliseconds to seconds [59,66]. As a restrictive factor, it should be noted that a prolonged membrane hyperpolarization of −110 mV or beyond endangers the vitality of neurons in acute brain slice preparations [67]. As in previous studies [27,30], we used a protocol with variable voltage-step durations, given the fact that HCN channel activation accelerates with increasing hyperpolarization [59]. However, our recordings might underestimate some effects, as we potentially did not reach a complete steady-state equilibrium during strongly hyperpolarizing voltage steps.

V_1/2_ values for HCN2 and HCN4 range between −70 and −100 mV [58] and, in the case of both isoforms, their steady-state activation is readily modulated by cAMP. The intracellular addition of cAMP can shift the V_1/2_ 10 to 25 mV toward depolarization [68]. In this study, TC neurons of the VB complex displayed a median V_1/2_ of −86.6 mV, as well as a median τ_slow_ of 1143 ms. Together with the existing research evidence, our data strongly indicate that the recorded conductance stems from HCN2 and HCN4, whereas the other studies cited include recordings of HCN1 and HCN2, HCN1, HCN2 and HCN3, and HCN2 andHCN3. It is further noteworthy that the DEX-induced inhibition of I_h_ in slice recordings of rat hypothalamic PVN neurons was almost completely antagonized by the application of yohimbine, an α_2_-adrenoceptor antagonist. This suggests that the inhibition of I_h_ was reliant on an intact α_2_-adrenoceptor mechanism [48]. 

The non-inactivating, inwardly rectifying current I_h_ plays a crucial role in the generation, amplification, and stable maintenance of rhythmic oscillations within the TC network [69] and further controls the resting membrane potential of TC relay neurons [41]. The intrinsic properties of thalamic HCN channels and their finely tuned interplay with other conductances, such as the low-threshold Ca^2+^ current I_T_ [21], promote the generation of oscillations in the delta frequency band, both in individual TC relay neurons [70], and within the TC network [71]. EEG recordings of patients receiving DEX sedation are characterized by slow delta oscillations superimposed with, at lower concentrations, sleep spindles, and this state is further distinguished by a noteworthy absence of profound unconsciousness [3,4]. Interestingly, these molecular actions converge on a preferential disruption of TC functional connectivity, while corticocortical functional connectivity was largely maintained [18]. Both EEG hallmarks of DEX sedation, slow delta oscillations and dose-dependent occurrence of spindle activity, are thought to originate within TC loop circuits [5]. This is further supported by the observation that DEX administration sufficient for a sedative state induces high power in the δ frequency band in local field potentials from thalamic nuclei. However, compared with the first-order VB complex, the higher-order central medial thalamus seems to be preferentially affected [20]. As DEX administration results in sedation and patients remain cooperative or can be easily aroused [72], TC interaction, especially communication between thalamic first-order nuclei of the VB complex (which are the focus of this study) and the primary somatosensory cortex, must preserve a certain degree of intactness. If DEX would indeed be able to strongly inhibit HCN channel function in TC relay neurons, with subsequent changes of biophysical membrane properties, then one would rather expect a profound disruption of sensory information relaying. Moreover, it has been demonstrated that DEX-induced sedation and hypnosis in mice are primarily mediated by the α_2A_-adrenergic receptor subtype [13,14,73]. Most pharmacological and physiological functions of α_2_-adrenergic receptors are ascribed to this specific subtype [15]. Interestingly, thalamic nuclei appear to primarily (albeit lightly) express the α_2B_-adrenergic receptor subtype [74]. Therefore, our observation that DEX does not potently inhibit thalamic HCN channel function is consistent with previous studies demonstrating the importance of α_2A_-adrenergic receptors in DEX-mediated sedation.

Naturally, the findings of this study should be interpreted in light of its limitations. Firstly, all recordings in this study were performed at room temperature. Albeit not uncommon [27,28,30,31,32,47,48], there is evidence that HCN channel activation and deactivation time constants significantly accelerate with increasing temperatures, whereas HCN channel-mediated current amplitudes were either not significantly changed by the temperature or the effect of temperature adjustments was much lower [75,76]. Further, all animals used in this study were female. There is limited evidence that in certain neuronal cell populations, HCN-mediated I_h_ might be regulated by circulating estradiol in a cyclical manner [77], though this does not appear to be a general concern so far [60,67]. However, sex-specific differences in HCN channel modulation should be more adequately addressed in future studies.

Ultimately, our data demonstrate that DEX, in a clinically relevant concentration range, does not significantly inhibit the native HCN channel function of TC neurons in the thalamic VB complex ex vivo, and parameters of neuronal excitability remain largely unaffected by its application. In comparison with anesthetic agents such as propofol, xenon, halothane, and sevoflurane, which substantially inhibit native HCN channel function in TC relay neurons in vitro [25,27,29,30], DEX administration does not as potently disrupt thalamic neuronal activity. Consistent with our patch-clamp findings, DEX-mediated slow delta oscillations and spindles exhibit smaller amplitudes compared with propofol [4]. These differences on a molecular level, including a different effect on native HCN channel function, could be part of the explanation for why DEX administration results in a sedation-like state, while other anesthetic agents can induce profound unconsciousness.

## 4. Materials and Methods

### 4.1. TC Slice Preparation

Animal handling was reviewed and approved by the competent veterinary office (Munich, Germany). We used female C57Bl/6N mice (P28–P35), which were put in deep isoflurane anesthesia (as indicated by loss of righting reflex) before surgical brain extraction. Brain slices with a 350 µm-thickness were prepared according to a well-established protocol [78], which allows for reliable identification of the VB complex of the thalamus, using a vibratome (HM 650 V, Thermo Fisher Scientific, Walldorf, Germany). Upon removal, brains were rapidly transferred into the ice-cold artificial cerebrospinal fluid (aCSF) optimized for brain slice preparation, containing (in mM) 125.0 NaCl, 2.5 KCl, 1.25 NaH_2_PO_4_, 25.0 D-glucose, 25.0 NaHCO_3_, 6.0 MgCl_2_, and 0.5 CaCl_2_. During the cutting procedure, aCSF was continuously aerated with carbogen (95% O_2_, 5% CO_2_) to ensure oxygenation and a stable pH of approximately 7.4. Slices containing the region of interest (3–4 per brain) were then transferred into a storage chamber containing a carbogen-saturated standard aCSF (in mM: 125.0 NaCl, 2.5 KCl, 1.25 NaH_2_PO_4_, 25.0 D-glucose, 25.0 NaHCO_3_, 1.0 MgCl_2_, and 2.0 CaCl_2_) and incubated for a minimum of 30 min at 34 °C in a warm water bath, followed by an additional 30 min of recovery time at room temperature.

### 4.2. Experimental Setup and DEX Application

Individual slices were transferred into the recording chamber of the patch-clamp setup and mechanically fixated with a custom-made grid. During experiments, slices were continuously perfused with carbogen-saturated standard aCSF at a flow rate of 5–8 mL/min. Upon visual allocation of the thalamic VB (Zeiss, Oberkochen, Germany), individual TC relay neurons were identified with infrared videomicroscopy (Hamamatsu Photonics, Herrsching, Germany). We established whole-cell recordings of single cells with fire-polished, borosilicate glass pipettes (Sutter Instrument, Novato, CA, USA). Pipettes were fabricated with open tip resistances between 4 and 6 MΩ, using a Flaming/Brown micropipette puller (P-1000, Sutter Instrument, Novato, CA, USA) and filled with an intracellular solution with the following composition (in mM): 130.0 K-D-gluconate, 5.0 NaCl, 2.0 MgCl_2_, 10.0 HEPES, 0.5 EGTA, 2.0 K_2_-ATP, 0.3 Na2-GTP, and pH: 7.25. As an electrophysiological defining feature, TC relay neurons exhibit an I_h_-dependent inward rectification of the membrane potential following injection of hyperpolarizing currents, termed voltage sag [40], which we used for the identification of the targeted neurons. After ensuring stable baseline recording conditions for at least 15 min, DEX was added to the aerated reservoir to attain three effective concentrations: 1 µM, 10 µM, and 100 µM. For the preparation of stock solutions, DEX hydrochloride was purchased from Sigma-Aldrich (Steinheim, Germany) and dissolved in aqua dest. for stock concentrations of 1 mM and 10 mM, whereas for the 100 µM concentration, DEX was directly dissolved in the aCSF of the circulation reservoir for each experiment, due to the solubility limitations of this concentration. Every brain slice was exclusively exposed to one DEX concentration. In order to ensure sufficient pharmacological slice penetration, we allowed for 30 min of DEX exposure before repeating recording protocols. The second set of recordings was performed after an additional 15 min (45 min after the beginning of the exposure). For approximately half of the recordings, we further conducted a washout by circulating fresh carbogen-saturated aCSF through the recording chamber for about 30 min before repeating the recordings. All experiments were recorded at room temperature (21–24 °C). All salts and chemicals used in these experiments were acquired from Sigma-Aldrich (Steinheim, Germany), with the exception of ZD7288 (Tocris, Wiesbaden-Nordenstadt, Germany).

### 4.3. Electrophysiology

All patch-clamp recordings were conducted in the whole-cell configuration with a discontinuous single-electrode voltage-clamp (dSEVC) amplifier (SEC 10L, NPI Electronic, Tamm, Germany). Though series resistance is effectively compensated by applying the dSEVC technique with high switching frequencies (60–80 kHz and a 25% duty cycle), it was continuously monitored. I_h_ currents were recorded in the voltage-clamp mode, according to a protocol previously described [27,30]. Starting from a membrane potential of −43 mV, I_h_ currents of TC neurons were gradually activated by applying hyperpolarizing voltage steps (10 voltage steps of 10 mV each, up to a maximum of −133 mV). As the stability of whole-cell recordings tends to deteriorate at highly hyperpolarized membrane potentials and factoring in the concurrent acceleration of I_h_ activation kinetics through hyperpolarization, we opted for incrementally reducing the pulse length of each voltage step by 500 ms, resulting in a 2.0 s pulse duration at −133 mV [29,41]. I_h_ appears to reliably reach its maximum at membrane potentials beyond −110 mV [79]. Therefore, maximal I_h_ current amplitude was derived from the −133 mV voltage step and used for normalization. Each voltage step also included a tail step of −103 mV before returning to the baseline of −43 mV. Respective tail current amplitudes (I_tail_) were normalized to the tail current acquired from the −133 mV voltage step and were used to assess the voltage dependency of steady-state activation. For this purpose, the equation (I–I_min_)/(I_max_–I_min_) was applied, with I_max_ representing the tail current amplitude derived from the voltage step of −133 mV to −103 mV and I_min_ derived from the voltage step from −43 mV to −103 mV. These data fit well with a Boltzmann distribution, which is usually applied to estimate steady-state activation. The voltage-dependent time course of I_h_ activation was approximated by fitting a biexponential function to the currents evoked during the hyperpolarizing step of −133 mV, yielding fast (τ_fast_) and slow (τ_slow_) time constants of I_h_ activation, as previously described for TC relay neurons [80]. Fitting with two exponentials is known to improve fits for a majority of I_h_ currents evoked at strongly hyperpolarized membrane potentials, while a monoexponential fit approach yields more robust results at more depolarized membrane potentials [81]. As we did not analyze time course activation as a function of membrane potential, we opted for applying a biexponential fit.

Further current-clamp recordings were recorded from the same TC neurons. After injection of a hyperpolarizing current of −350 pA for a duration of 500 ms at the resting membrane potential, TC neurons display an inwardly directed rectification of the membrane potential, termed voltage sag. We defined the voltage sag as the difference between the maximum voltage deflection and the steady-state potential. Subsequently, we measured the time between the start of the membrane repolarization at the end of the hyperpolarizing current injection and the peak of the first action potential during rebound burst-firing (rebound burst delay). This current-clamp protocol was repeated eight times for each recording and later analyzed as a graphically averaged composite trace with Igor 5 (WaveMetrics, Lake Oswego, OR, USA). The impact of DEX on active and passive electrical membrane properties, comprising resting membrane potential, membrane input resistance, the current–voltage relationship, the threshold value of action potential firing, and the tonic action potential frequency were derived from membrane voltage changes induced by successive increases of current injections (starting from −90 pA to +360 pA, 10 pA increase per injection with a 500 ms duration) at the resting membrane potential. For current-clamp recordings, we only adjusted the resting membrane to control (pre-drug application) levels when significant changes in the resting membrane potential were observed.

The liquid junction potential was corrected online, as estimated based on the ionic compositions of intra- and extra-cellular solutions and with help of the Liquid Junction Potential Calculator tool (Clampex 10.7, Molecular Devices. San Jose, CA, USA). As previously elaborated on ([27,30], all experiments were recorded in presence of 0.15 mM barium chloride (Ba^2+^), which was added to the aCSF circulation. Briefly summarized, Ba^2+^ inhibits the potentially interfering effects on inwardly rectifying potassium channels (K_ir_) and members of the two-pore domain acid-sensitive potassium channels (K_2P_), as well as facilitates improved analysis by significantly enhancing voltage sag amplitude [40].

The current responses were amplified, low-pass filtered at 3 kHz, digitized (ITC-16 Computer Interface, Instrutech Corporation, Great Neck, NY, USA) with a sampling frequency of 9 kHz, and stored on a hard drive (HP ProDesk 400 G2.5 SFF, Hewlett-Packard, Palo Alto, CA, USA). For data acquisition, we used Pulse version 8.5 (HEKA Elektronik, Lambrecht, Germany). Data from individual cells were only included in the analysis if they did not meet one of the following exclusion criteria: (1) no GΩ-seal resistance formation upon establishment of cell-attached configuration, (2) an initial resting membrane potential below −50 mV, (3) an unstable series resistance or holding current (>20% change), and (4) no stable baseline conditions after 15 min, as assessed by the experimenter. Analysis of electrophysiological recordings was performed offline with Igor 5 (WaveMetrics, Lake Oswego, OR, USA). Action potentials were evaluated using Mini Analysis (Synaptosoft, Fort Lee, NJ, USA).

### 4.4. Statistical Analysis

For statistical evaluation of the data obtained and the creation of graphs, we used GraphPad Prism 7.03 (GraphPad Software, San Diego, CA, USA). Sample sizes for different concentration groups were chosen based on previous experience. As formally testing for normal distribution inherently poses a challenge in case of limited sample sizes [82], we opted for the application of non-parametrical testing by default. Accordingly, all results are presented as medians with the interquartile range (IQR). We used the Kruskal–Wallis test for independent groups with Dunn’s posthoc test to correct for multiple comparisons. In the first step, we tested for significant differences between the medians of the control values of the three different concentration groups to ensure comparability between groups. In the second step, we tested for significant differences between absolute changes of the three concentration groups compared to a normalized control. For multiple comparisons of repeated measures, we used the non-parametric Friedman test. In the case of simple comparisons, we used the Wilcoxon matched-pairs signed-rank test. For sample sizes with n ≤ 5, we only provide a descriptive statistic, as non-parametric statistical tests cannot be meaningfully applied in these cases. Where applicable, *p*-values were adjusted for multiplicity and differences were considered statistically significant if *p* < 0.05, as graphically indicated by asterisks.

## Figures and Tables

**Figure 1 ijms-24-00519-f001:**
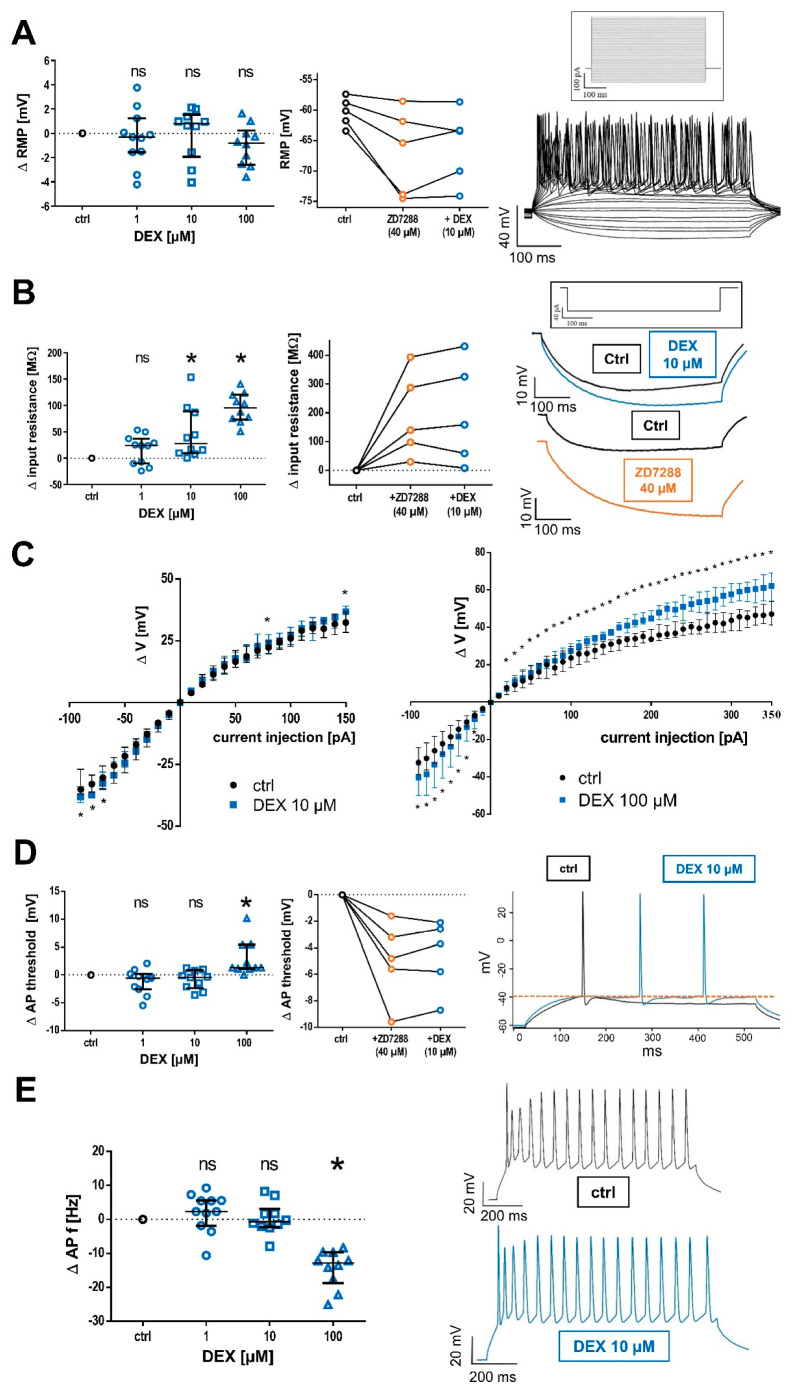
(**A**) DEX did not affect the resting membrane potential (RMP, control: −58.2 [−59.7–−56.9] mV) over the concentration range from 1 µM to 100 µM after 45 min of exposition, as seen in the left panel. In contrast, the HCN channel blocker ZD7288 (40 µM) hyperpolarized every cell exposed (n = 5) and this effect persisted after coapplication with 10 µM DEX, as seen in the middle panel. For analysis of active and passive biophysical membrane properties, currents from −90 pA to +360 pA (increments of 10 pA) were injected into TC relay neurons, with the corresponding current protocol shown above. On the right, representative voltage traces under the baseline conditions are recorded in the current-clamp mode (for a better overview, only every third voltage trace is included). The RMP was derived from the 0 pA current injection step. Δ: absolute change compared to control and ns: not significant. (**B**) DEX significantly increased neuronal input resistance at a hyperpolarizing current injection of −90 pA, beginning at a concentration of 10 µM and indicating a decrease in cell leakiness during membrane hyperpolarization. Control: 343.4 (252.3–365.6) MΩ, in the left panel. In the middle panel, the effects of ZD7288 on native recordings and after coapplication with 10 µM DEX are shown. ZD7288 induced a strong increase in neuronal input resistance in all cells. Representative voltage traces under baseline conditions and in the presence of 10 µM DEX (blue), as well as 40 µM ZD7288 (orange), are depicted on the right. The input resistance was derived from the voltage changes following a current injection of −90 pA. The accompanying current protocol is shown above and * *p* < 0.05. (**C**) Current-voltage relationships were created by injecting currents from −90 pA to +360 pA (10 pA increments) into TC neurons and the change (Δ V) in membrane potential was recorded. The resting membrane potential was derived from Δ V = 0 mV. Significant increases of Δ V were only reliably observable in presence of 100 µM DEX over the complete current range, whereas the effects of 10 µM DEX were restricted to strongly hyperpolarizing current injections. (**D**) A total of 100 µM DEX shifted the voltage threshold for action potential (AP) generation toward depolarization, while the threshold remained unchanged at lower concentrations. The AP threshold under control conditions was −39.2 (−40.5–−37.6) mV (left panel). In contrast, ZD7288 shifted the AP threshold in all recorded cells toward hyperpolarization, as shown in the middle panel. Representative voltage traces under control conditions and in the presence of 10 µM DEX are depicted on the right. The threshold is indicated by a dotted line. (**E**) Similarly, only the concentration of 100 µM DEX significantly decreased the frequency (f) of tonic action potential (AP) firing, induced by depolarizing current injections. Corresponding voltage traces under control conditions and after the application of 10 µM DEX are shown on the right.

**Figure 2 ijms-24-00519-f002:**
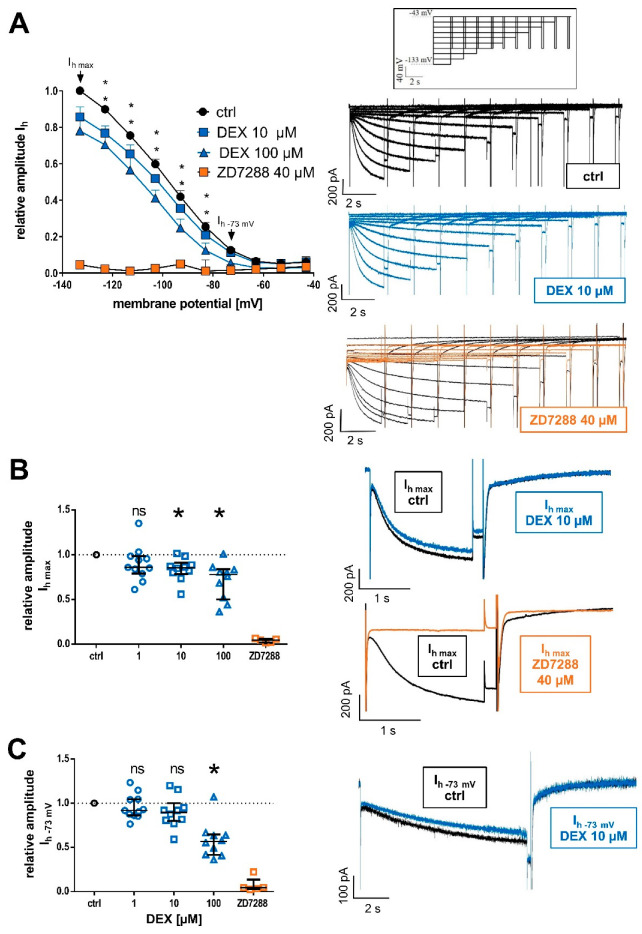
(**A**) Relative changes of HCN-mediated I_h_ current amplitudes at respective membrane potentials after DEX application (10 µM and 100 µM) compared to the control and the effect of the selective HCN channel blocker ZD7288 (40 µM). Beginning at −83 mV, 10 µM DEX induced a significant reduction in I_h_. On the right, representative current traces from a TC relay neuron under control conditions and after the application of 10 µM DEX for 45 min. In comparison, 40 µM of ZD7288 resulted in a near complete attenuation of I_h_ currents over the complete voltage range. The voltage-clamp protocol is shown above and * *p* < 0.05. (**B**) Relative reduction in the maximal I_h_ current amplitude (I_h max_) was recorded when the cell was hyperpolarized to −133 mV. The observed changes were significant beginning at a concentration of 10 µM DEX. On the right, representative current traces of I_h max_ under control conditions and in the presence of 10 µM DEX. The reduction observed after adding 40 µM of ZD7288 (orange) was almost complete, as depicted below. (**C**) At physiological membrane potentials (−73 mV), the reduction in relative I_h_ amplitude was only significant after the application of 100 µM DEX. HCN channel blocker ZD7288 reduced I_h−73 mV_ almost completely in all cells (n = 5). On the right, representative current traces of I_h_ at −73 mV under control conditions and in the presence of 10 µM DEX. Ns: not significant.

**Figure 3 ijms-24-00519-f003:**
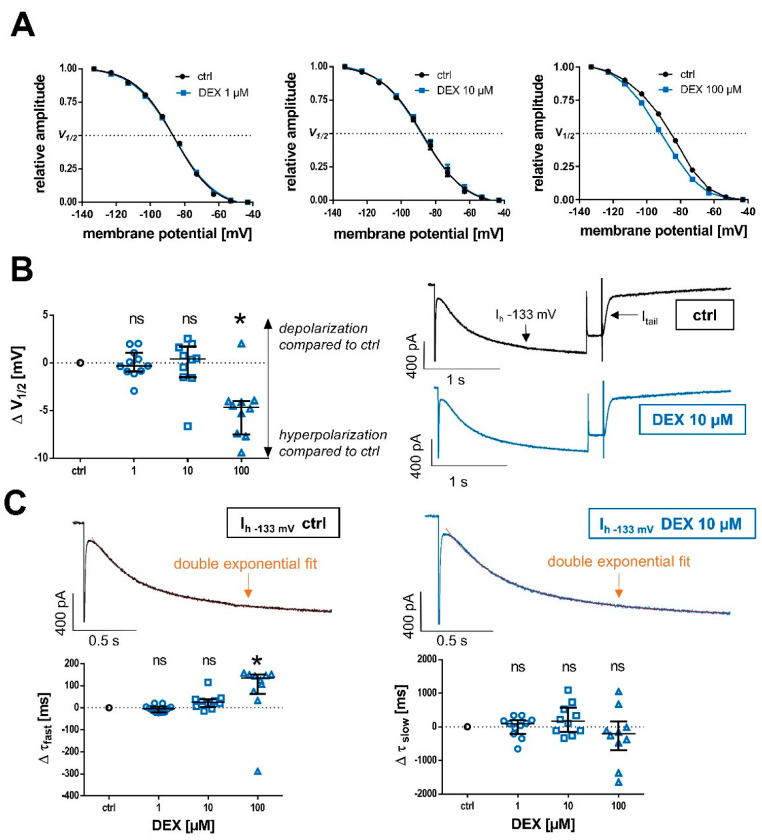
(**A**) In order to analyze voltage-dependent channel gating and steady-state activation curves of HCN channels under control conditions and after the application of increasing DEX concentrations, normalized tail current amplitudes (I_tail_) were fitted to a curve using the Boltzmann function. The tail step was set at −103 mV for each voltage step before returning to −43 mV. V_1/2_: half-maximal activation voltage. (**B**) Only the high concentration of 100 µM DEX led to a significant left shift (toward hyperpolarization) in V_1/2_ compared to the control (−86.6 [−85.2–−88.0] mV). Respective current traces under control conditions and after the application of 10 µM DEX are depicted on the right. Tail currents (I_tail_) were measured subsequent to the hyperpolarizing voltage step at the fixed “tail” step of −103 mV before returning to −43 mV. Δ: absolute change compared to control. Ns: not significant. * *p* < 0.05. (**C**) Representative current traces of I_h_ at the hyperpolarizing voltage step of −133 mV under control conditions and after 45 min of DEX application (10 µM) are depicted above. By fitting a biexponential function to I_h_ (as indicated by the dotted line), fast (τ_fast_), and slow (τ_slow_) time constants of time-dependent activation were determined. Only the high concentration of 100 µM DEX resulted in a significant deceleration of the fast activation constant compared to the control (231 [214–273] ms), presented as absolute changes. The slow time constant was not affected (control: 1143 [986–1405] ms).

**Figure 4 ijms-24-00519-f004:**
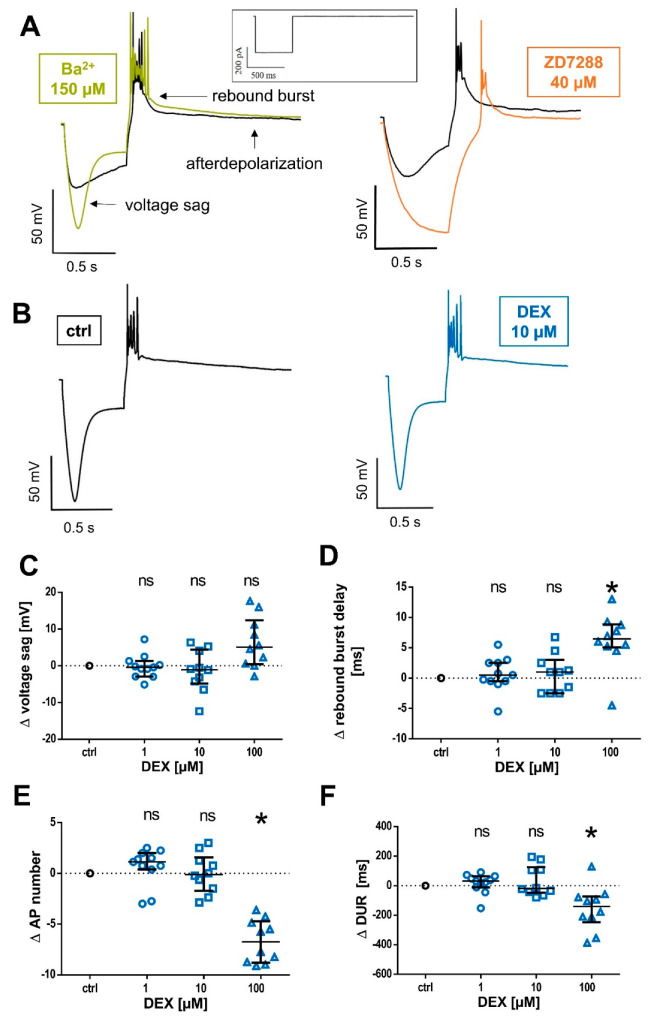
(**A**) TC replay neurons display an inwardly directed rectification of the membrane potential upon hyperpolarizing current injection (−350 pA, 500 ms, and the current protocol is shown on top), termed voltage sag. The voltage sag is unmasked in the presence of Ba^2+^ (green, 150 µM), on the left panel. Low-threshold calcium spike rebound burst-firing (rebound burst) is followed by an HCN channel-mediated afterdepolarization. In comparison, there is no quantifiable voltage sag in the presence of HCN channel blocker ZD7288 (orange, 40 µM), corresponding to a complete suppression. (**B**) Exemplar current-clamp recordings in the absence and after 45 min of 10 µM DEX (blue) administration with no significant changes. (**C**) DEX had no significant effect on voltage sag amplitude compared to the control (86.4 [70.7–92.9] mV). Δ: absolute change compared to the control. Ns: not significant. (**D**) Compared to control measurements (29.0 [27.0–30.8] ms), the delay of rebound burst-firing was only prolongated in the presence of 100 µM DEX. * *p* < 0.05. (**E**) The median number of action potentials (AP) during rebound burst spiking (control: 9.5 [7.3–12.5]) was only significantly reduced after the application of 100 µM DEX. (**F**) Accordingly, only the high concentration of 100 µM DEX significantly abbreviated the median rebound burst duration (DUR) compared to the control (225.5 [126.5–399.0] ms).

## Data Availability

The raw data supporting the conclusions of this article will be made available by the corresponding author upon reasonable request, without undue reservation.

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
