# Peer review of "Sedative Properties of Dexmedetomidine Are Mediated Independently from Native Thalamic Hyperpolarization-Activated Cyclic Nucleotide-Gated Channel Function at Clinically Relevant Concentrations"

_ijms, 2022, doi:10.3390/ijms24010519_

Round 1
Reviewer 1 Report
In this paper, the authors investigated the effect of Dexmedetomidine on Ih inhibition in thalamocortical relay neurons of the ventrobasal complex of the thalamus at clinical concentration (1-10 microM). Patch-clamp recordings in voltage-clamp and current-clamp mode were performed. They found that at 10 microM, DEX reduced maximal Ih amplitude and increased input resistance, but there was no effect on another Ih properties. The results of this paper provide an insight about the neuronal properties related to the sedative effect of DEX.
In my opinion, patch-clamp recordings were well executed and the conclusions are well supported by the data. However, I have certain concerns that I will mention below:
Minor points:
11) In section 2.1, only current-voltage curves were done, but the term “voltage-current” appears multiple times (lines 121, 126, 131).
22) In Fig 1B, the legend mentions an orange trace corresponding to ZD7288 application, but there is no such a trace in the Figure.
33) In Fig 3C, in control plots, the Y axis title is missing.
44) Add mM after 0.15 in line 640.
Author Response
We want to thank the reviewer for the favorable review.
1) The reviewer is correct in pointing out this mistake. We changed "voltage-current" in the corresponding lines to "current-voltage".
2) We updated Figure 1 B, which now includes the ZD 7288 voltage trace (orange) as described in the figure legend. Concurrently, we also revised Figure 2 B, where we noticed a similar omission. It now includes the depiction of an Ih max current trace after the application of ZD 7288, as (already previously) described in the figure legend.
3) We updated Figure 3 C, which now includes the missing Y axis title (Δ τfast [ms])
4) We included the missing "mM" after 0.15 in line 640 and thank the reviewer for noticing.
With kind regards,
Stefan Schwerin, for the authors
Reviewer 2 Report
The manuscript titled “Sedative properties of dexmedetomidine are mediated independently from native mice thalamic hyperpolarization-activated, cyclic-nucleotide gated channel function at clinically relevant concentrations” from Schwerin et al. uses whole cell patch clamp recordings to study the effect of dexmedetomidine (DEX), a medication clinically used for sedation, on native HCN channel function of thalamocortical relay neurons in the ventrobasal complex of the thalamus from mice. Based on recent findings which suggest that DEX might act on HCN channels in dorsal root ganglia neurons and on the authors’ and other groups’ prior findings that a number of other α2-adrenoceptor agonists can inhibit HCN channel functions, the authors investigated here whether DEX also affects native HCN channel function and HCN channel-mediated TC relay neuron excitability. The authors found that under clinically relevant concentrations of DEX, however, the modification of biophysical membrane properties and HCN-channel mediated neuronal excitability is modest. Therefore, DEX does not seem to inhibit the native HCN channel function of TC neurons significantly. The experiments were reasonably designed, and the paper was logically written. The author also thoroughly discussed the results in the discussion section. I do not have significant concerns about the article. Here are some minor suggestions:
In Figure 1 B, line 176, the figure legend says “Representative current traces under baseline conditions and in the presence of 10 µM DEX (blue) as well as 40 µM ZD7288 (orange) are depicted on the right.” However, in Figure 1 B, right panel, there is no trace shown in orange for ZD7299. Please add the missing trace for ZD7288. In addition, the traces of control(black) and DEX(blue) shown in this panel appear to be voltage traces. However, in the figure legend, they are described as current traces. Please check.
Line 80, “…HCN channel function in neurons form dorsal root ganglia [31,32].” Check if the authors mean “… in neurons from dorsal root ganglia”
Line 192, “2.2. Influence of DEX on HCN channel function an Ih-currents”, it appears “an” should be changed to “and”
Author Response
We want to thank the reviewer for the favorable review.
1) We updated Figure 1 B, which now includes the missing depiction of the ZD7288 voltage trace. We further corrected the corresponding figure legend and changed "current traces" to "voltage traces", as the reviewer rightly noticed. Unfortunately, we noticed a similar mistake in Figure 2 B, which we also revised. It now includes the depiction of a ZD7288 current trace (Ih max), as described in the corresponding figure legend.
2) We thank the reviewer for noticing this spelling error, the sentence was changed accordingly into "attenuate HCN channel function in neurons from dorsal root ganglia"
3) We also changed the heading in (formerly) line 192 (now 195), it now reads "... HCN channel function and Ih-currents"
With kind regards,
Stefan Schwerin, for the authors